# IQA-Octopus: Unified Multi-Granularity Image Quality Assessment with Reasoning, Grounding and Referring

## Abstract

We present IQA-Octopus, the first image quality assessment (IQA) framework that unifies reasoning, grounding, and referring. Built upon large multi-modality models (LMMs), IQA-Octopus is designed to perform multi-functional quality analysis by enabling multi-granularity perception capabilities. Existing LMM-based IQA models can merely support partial perception dimensions, *e.g.*, quality descriptions and question answering (*i.e.*, reasoning), pixel-wise grounding individually due to the lack of (i) unified IQA datasets with multi-functional annotations and (ii) a proper optimization paradigm assisting multi-granularity perception. To overcome this, we built the first multi-functional datasets incorporating global/local reasoning, pixel-wise grounding, and region-wise referring tasks, followed by an elaborately designed automatic multi-granularity dataset extension method, which ensures the optimization of our IQA-Octopus. To facilitate multi-functional perception, we introduce a conflict-free two-stage optimization strategy that progressively transfers multi-granularity textual understanding capability to pixel-wise perception: (i) The first stage injects the textual multi-granularity perception into IQA-Octopus with the joint optimization of multiple textual-based perception tasks, including reasoning and referring, and (ii) The second stage introduces a novel text-to-point strategy in the pixel-wise perception stage, which implicitly warps the text logits to coordinates of pixel-wise grounding in a zero-shot manner. Based on the above two efforts, we achieve our IQA-Octopus by unifying multi-functional and multi-granularity explainable image quality assessment. Our model achieves comparable or state-of-the-art performance across multiple benchmarks with limited training data, demonstrating strong multi-granularity understanding and remarkable versatility. The code and dataset will be released after acceptance.

## 1 Introduction

As a fundamental task in the visual signal processing field, image quality assessment (IQA) aims to evaluate the quality of visual signals by modeling human visual system (HVS). With the rapid growth of streaming services and the explosive increase in user data, particularly images, there is a growing demand for more explainable and powerful IQA methods to better guide image compression, image restoration, and preference optimization for AIGC content, in order to enhance user quality experience. Recently, the rapid development of Large Multi-modality Models (LMMs) boosts the advancement of IQA. These LMM-based IQA methods have improved in both score rating accuracy and explainability. Q-Bench (Wu et al., 2023a) first proposed a benchmark to evaluate the perceptual quality ability of LMMs. Q-Align (Wu et al., 2023b) demonstrates highly promising results for visual scoring. Some other works (Wu et al., 2024a; You et al., 2024b; Lu et al., 2025) unlock the quality description ability of LMMs. However, these works solely focus on the global quality evaluation, as observed in DepictQA-Wild (You et al., 2024b), the LMM-generated quality description fails to point out the distortion location, revealing their poor fine-grained ability. Although Q-Ground (Chen et al., 2024a) endows the model with quality grounding capability, it is not generalizable to other quality assessment tasks. Moreover, its applicability is limited to distortion-type grounding, covering only a narrow set of distortion categories.

To achieve comprehensive global and fine-grained quality perception ability, we propose a unified multi-granularity dataset **IQA-Octopus-33K** that draws special focus to region-wise quality, which

includes four task paradigms: **(1) Global Quality Description**, **(2) Local Quality Description**, **(3) Visual Quality Grounding**, **(4) Visual Quality Referring**. Compared to previous datasets (Wu et al., 2024a; Chen et al., 2024a;c) for explainable image quality assessment, our dataset offers the following key advantages: i) **Comprehensive Task Taxonomy**: For Visual Quality Grounding and Referring, we decompose the tasks into a set of structured sub-tasks to address the diverse requirements of different application scenarios. To address the limited distortion diversity in existing datasets, we further define a comprehensive degradation space that encompasses both synthetic and in-the-wild distortions, thereby ensuring broader distortion coverage. ii) **Efficient and Scalable Annotation Pipeline**: We employ a hybrid labeling approach using SAM-based tools and the open-source LMM (*i.e.*, InternVL-2.5 (Chen et al., 2024b)), enabling fully automatic annotation for synthetic distortions and semi-automatic labeling for authentic distortions. iii) **Enhanced Data Scale through Integration**: To further increase diversity and training scale, we integrate widely-used datasets such as Q-Instruct (Wu et al., 2024a) and DQ-495K (You et al., 2024b). These datasets enable our model to undergo a two-phase training process, progressively advancing from text-based multi-granularity understanding to pixel-wise comprehension. In the first phase, the model is instruction-tuned to be fine-grained and can tackle all kinds of text-based quality tasks in multi-granularity. In the second phase, we enhance the model's capability for pixel-wise quality grounding by integrating a dedicated grounding head(*e.g.*, Segment Anything Model (Kirillov et al., 2023)). Existing approaches such as Q-Ground (Chen et al., 2024a) and other LMM-based grounding methods (Lai et al., 2024; Rasheed et al., 2024) adopt explicit special token (*i.e.*, $<seg>$) generation to interface with the grounding head. However, on the LMM side, this practice has been shown to impair instruction-following behavior and disrupt reasoning processes (Wu et al., 2024c), primarily due to the inclusion of the special segmentation tokens in the text vocabulary. On the grounding head side, it also necessitates additional fine-tuning to effectively handle the special token. To simultaneously achieve region-wise referring and pixel-wise grounding without degrading the model's reasoning ability, we implement a training-free grounding approach by proposing a text-to-point strategy for prompt generation. Specifically, we keep the text output stainless and implicitly convert text logits into point coordinates that reflects the location of the target. We bridge the gap between LMMs and universal segmentation models in a simple yet effective way, making our approach plug-and-play for any LMM. Our approach combines multi-granularity quality assessment tasks and presents a novel strategy that enables grounding in a training-free manner, offering greater potential for real-world applications. The main contributions are as follows:

- We present a new dataset, **IQA-Octopus-33K**, that opens opportunities for training model with multi-granularity perception. To the best of our knowledge, we are the first to formally define the local quality description task for a fine-grained challenge. We reconsider visual quality grounding and referring task with sophisticated design, making it more challenging yet better aligned with real-world applications.

- We propose a new LMM-based multi-functional model with conflict-free two-stage optimization. Moreover, we introduce a training-free text-to-point grounding strategy for LMM-based grounding, enabling grounding without the loss of reasoning ability. Extensive experiments validate that our method outperforms the existing models in our various datasets.

- We address the importance of multi-granularity perception ability and establish new benchmark for text-based quality question answering and pixel-wise quality grounding. Our model demonstrates state-of-the-art performance in three major task categories: text-based answering, pixel-wise grounding and visual quality scoring. These results are substantiated through rigorous evaluations conducted on multiple benchmark datasets, including our established benchmark, Q-Bench-A1 and Q-Ground-Test, which further validate the comprehensive quality understanding capability and versatility of our model.

## 2 RELATED WORK

**LMM-based Image Quality Assessment.** The LMM-based IQA models have progressed along three dominant pathways. The first (Zhang et al., 2023b; Wang et al., 2023; Wu et al., 2023b; You et al., 2025) focuses on feature-text alignment, where LMMs are adapted to map visual quality characteristics into textual representations. The second direction (Zhu et al., 2024; Wu et al., 2024d)

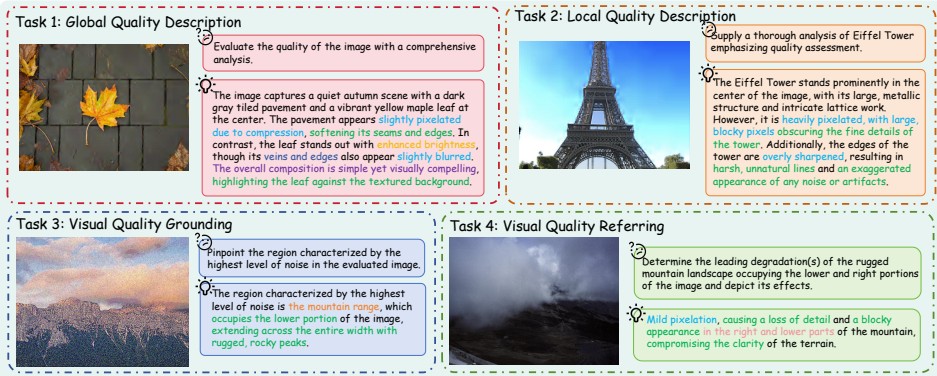

Figure 1: **The components of our task paradigms**, include (a) Global Quality Description, (b) Local Quality Description, (c) Visual Quality Grounding, and (d)Visual Quality Referring.

explores specialized prompt strategies, which are designed to unlock the inherent quality perception ability of LMM. The third pathway advances explainable IQA (EIQA), especially with the integrated capability of LMMs. Early works introduce a low-level benchmark (Wu et al., 2023a) for general LMMs (Zhang et al., 2023a; Zhu et al., 2023; Chen et al., 2023b; Gao et al., 2023) and several quality-specific instruction tuning datasets (Wu et al., 2024a; You et al., 2024b; Wu et al., 2024b), focusing on both holistic quality and attribute-specific perception. To enable finer-grained understanding, Q-Ground (Chen et al., 2024a) conducts pixel-wise quality grounding but compromises instruction-following and reasoning. Grounding-IQA (Chen et al., 2024c) adds quality referring via bounding boxes, but lacks general applicability. To address more realistic and diverse scenarios, we revisit these paradigms and propose a multi-task dataset supporting quality assessment from global to pixel level, without degrading chat capabilities.

**LMM-based Visual Grounding.** LMM-based grounding models can localize objects with complex reasoning during user interactions. Some methods (Bai et al., 2025; Chen et al., 2023b; Peng et al., 2023; You et al., 2023) output bounding box coordinates through textual token generation. Text4Seg (Lan et al., 2024) casts image segmentation as text generation, which fails in dense scenarios. In contrast, several recent approaches (Lai et al., 2024; Rasheed et al., 2024; Wei et al., 2024; Xia et al., 2024) train LMMs to predict a special segmentation token (*i.e.*, $<seg>$) that represents the grounded object and guide the segmentation head (*e.g.*, SAM (Kirillov et al., 2023)) to generate masks, which can lead to degradation in instruction-following ability. Unlike these methods, we perform grounding in a zero-shot manner by implicitly mapping the positional terms to point coordinates for the segmentation head. This eliminates the need for grounding instruction-tuning data while maintaining the model's original instruction-following capability.

## 3 TASK PARADIGMS AND DATASET CONSTRUCTION

### 3.1 TASK PARADIGMS

To enable multi-granularity perception, we propose a four-task paradigm with a primary focus on region-wise quality understanding, which is shown in Fig. 1. For better applicability, we conduct all tasks under the no-reference setting. The four tasks are introduced as follows:

- **Task 1: Global Quality Description.** Follow the existing works (You et al., 2024b; Wu et al., 2024a), we conduct this task by given a target image and output the quality-related answer. Specifically, our dataset emphasizes more detail(*e.g.*, texture damage) in this task.

- **Task 2: Local Quality Description.** This task requires the model to produce region-wise quality descriptions, which calls for localized perception to first identify relevant areas, and then assess their quality factors.

- **Task 3: Visual Quality Grounding.** For this task, given an image and a question, the model is required to first produce a textual region-wise answer, followed by a pixel-wise segmentation mask. Furthermore, we decompose the task into three sub-tasks, each defined by a distinct grounding objective. (a) hybrid distortion intensity grounding (HyD-G). The

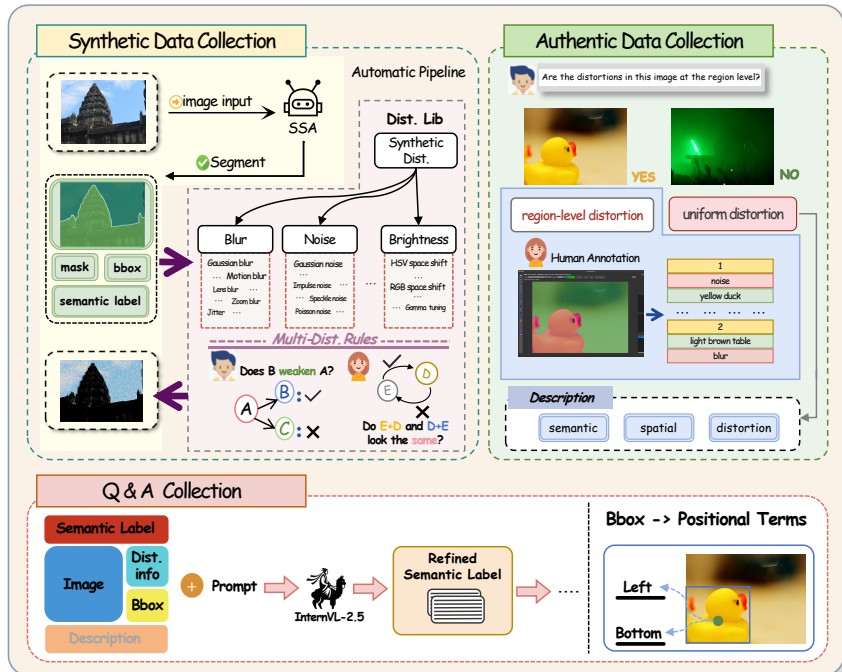

Figure 2: **Dataset construction pipeline.** For synthetically distorted data, we design a fully automatic pipeline that begins with SSA segmentation and proceeds with region-wise hybrid distortion accumulation. For authentically distorted data, we involve human annotators to identify and categorize region-level distortions. These data are then used as prompts or image inputs for InternVL-2.5 to generate question & answer pairs.

goal of this task is to ground the region with the most prominent (*i.e.*, maximum or minimum) cumulative intensity resulting from all types of present distortions. (b) single distortion intensity grounding (SiD-G). In contrast, this task focuses on a specific distortion within all present distortions, aiming to ground the region with the highest or lowest intensity of that particular distortion in the image. (c) distortion accumulation order grounding (DAO-G). Given a predefined distortion accumulation order, the objective of this task is to identify the region that matches both the distortion types and their accumulation order.

- **Task 4: Visual Quality Referring.** This task focuses on identifying distortion types within referred image regions. To support both concise and detailed responses, we design two answer patterns of increasing complexity. The *short-answer* pattern requires the model to accurately identify all types of distortions present in the region. The *long-answer* pattern extends this by also requiring the model to describe the perceptual effects of these distortions, enabling more comprehensive region-wise understanding.

## 3.2 DATASET CONSTRUCTION

Following our task paradigms, we construct a unified multi-granularity dataset, IQA-Octopus-33K, to support both quality reasoning and pixel-wise quality grounding, which comprises two subsets: 1) The instruction-tuning subset consists of {*image*, *question*, *answer*} triplets, enabling multi-level, text-based quality assessment in the first training stage. 2) The grounding subset augments these triplets with segmentation masks, supporting pixel-wise supervision in the second stage. To efficiently construct this dataset at scale, we adopt a hybrid annotation pipeline that leverages multiple open-source models. We first preprocess images using SAM-based tools and human annotators to identify distortion regions. Then, we employ InternVL-2.5(Chen et al., 2024b) to generate multi-turn question & answer pairs. The detailed construction pipeline is provided in Fig. 2.

**Image Data Collection.** In this stage, we collect all necessary image data and distortion-related annotations, including region-wise distorted images, distortion type and intensity, semantic region labels, segmentation masks, bounding boxes, and a brief quality description for each image. We take both authentic and synthetic distortions into consideration, which the original image data is

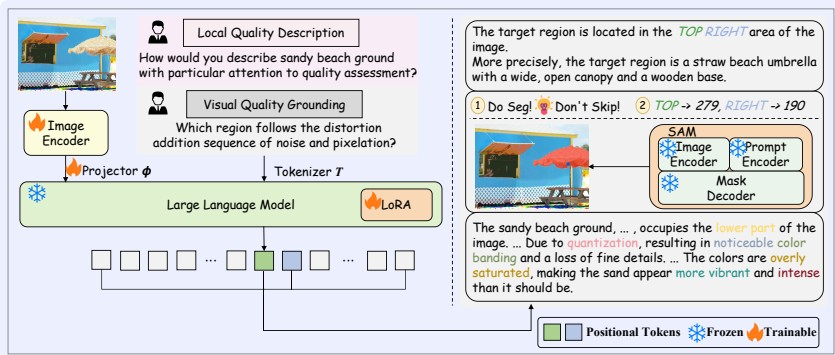

Figure 3: **Model framework.** An LMM with a segmentation head (*i.e.*,SAM) is employed for multi-granularity perception. Text-based reasoning guides whether to perform zero-shot segmentation, triggered only under non-uniform distortions.

chosen from KonIQ (Hosu et al., 2020) and KADIS-700K (Lin et al., 2019) dataset respectively. For images with authentic distortions—either uniformly distributed or localized at the region level—we employ human annotators to provide reliable ground-truth annotations. Under the human decision, the image with region-wise distortions is annotated with all the image data information, and the globally distorted image is solely required to a brief description. To ensure scalability and transparency in data construction, we deliberately avoid using expensive and closed-source large multimodal models (*e.g.*, GPT-4V (OpenAI, 2023)) for authentically distorted images.

For synthetic distortions, we propose an automated, human-free pipeline that generates region-wise distortion data from high-quality source images. Existing datasets are often limited in distortion diversity or only provide global-level degradations, which do not satisfy our needs for fine-grained region-wise quality assessment. To overcome this, we leverage the open-source tool **Semantic Segment Anything (SSA)** (Chen et al., 2023a) to extract semantic instance regions. We then apply various synthetic distortions to these regions, guided by the SSA-generated segmentation masks. This process automatically produces distorted images, region-wise distortion masks, distortion type labels, and corresponding semantic region labels—**all without human annotation**. To better simulate realistic degradation patterns, we adopt the multi-distortion composition strategy proposed in (You et al., 2024b), and manually define all perceptually recognizable distortion accumulation orders in accordance with the existing constraints of the human visual system. See more details in Appendix A.2.

**Question & Answer Collection.** We collect all questions and answers through an automated pipeline, which is composed of the following steps: **i) Defining the Question Format**: We first prompt InternVL-2.5 to define the structure of the question format. This ensures that all generated questions follow a consistent and standardized schema. **ii) Multi-round Dialogue for Answer Generation**: Once the question format is established, we initiate a multi-turn dialogue with InternVL-2.5 incorporating image information. Through this iterative dialogue, answers are progressively generated and filled into the predefined format. **iii) Semantic Label Refinement**: To prevent incorrect semantic labels initially generated by SSA, we refine and correct the semantic labels before proceeding to answer generation. This ensures higher semantic accuracy. **iv) Grounded Answer Generation**: For grounding-related answers, we generate both spatial descriptions and region-wise semantic descriptions using a set of predefined positional terms. Specifically, the terms *left* and *right* are used to indicate horizontal (x-axis) positions. And the terms *top* and *bottom* are used to indicate vertical (y-axis) positions. These positional terms are selected by converting the center of bounding box coordinates of the target object. These positional terms are mapped by the normalized coordinates of the center point of the bounding box of the target object, which is performed as the following equations:

$$T_x = \begin{cases} left, & \frac{x}{W} \in [0, \frac{1}{2}) \\ right, & \frac{x}{W} \in [\frac{1}{2}, 1] \end{cases} \qquad T_y = \begin{cases} top, & \frac{y}{H} \in [0, \frac{1}{2}) \\ bottom, & \frac{y}{H} \in [\frac{1}{2}, 1] \end{cases} \qquad (1)$$

## 4 METHODOLOGY

Our proposed model is designed for multi-granularity image quality perception through conversation, addressing tasks from global understanding to fine-grained region-wise referring and pixel-wise

Query : Provide a careful evaluation of the sky with attention to quality assessment.

Ours: The sky, a vast expanse of light blue, occupies the upper portion of the image. However, it is marred by a grainy texture due to noise, with small, random specks of color scattered throughout. Additionally, the sky appears slightly blurred, softening the edges and reducing the sharpness of the blue expanse.

DepictQA: The image depicts an urban railway yard with multiple trains, tracks, and buildings in the background. The sky is a clear blue with a few clouds. The evaluated image exhibits a slight noise distortion, which manifests as graininess throughout, particularly noticeable in the sky and on the train surfaces. This noise slightly reduces the clarity and sharpness of the image. Overall, the image maintains good quality with only a minor degradation due to the noise, preserving the integrity of the scene despite the distortion.

Q-Instruct: The sky in the image has rich colors, but the overall clarity is poor. The image has a significant amount of noise, resulting in poor quality.

Co-Instruct: The image has accurate exposure with no obvious overexposed or underexposed areas. The focus is precise, and the picture is clear with no noticeable noise. The colors are rich, and there is sufficient lighting. The texture details are clear, and the composition is centered. The main subject, the train in the middle, is clearly defined. Therefore, the quality is good.

(a) Visualization of Local Quality Description

Query : What region follows the pattern of adding pixelation before contrast strengthening in the evaluated image?

Ours answer: The target region is located in the *top left* area of the image. More precisely, the target region is a blurred background with indistinct shapes and colors.

| Input | Ground Truth | Ours | Q-Ground | Q-Instruct | LISA | Sa2VA |

(b) Visualization of Viusal Quality Grounding

Figure 4: **Visualization of different methods.** (a) Visualization of Local Quality Description task. (b) Visualization of the distortion accumulation order grounding (DAO-G) sub-task in Visual Quality Grounding.

grounding. To achieve this, we design a unified framework that progressively enhances the model's perception capabilities across different levels. The model first learns to understand image quality at the global-level and region level through instruction tuning, then escalates to pixel-wise grounding in a zero-shot manner via a novel text-to-point strategy.

## 4.1 OVERALL FRAMEWORK

Our framework consists of two main components: a LMM backbone and a segmentation head. An overview of our model is shown in Fig.3.

**LMM Backbone.** The LMM takes an image and a question as input and outputs a textual answer. The image is encoded and projected into the textual embedding space, while the question is tokenized and embedded. Both are then fed into a LLM to generate the answer. To improve efficiency and reduce redundant computation, the model determines the necessity of grounding based on the generated textual answer. If the textual answer indicates that the target region covers the entire image, grounding is skipped and the full image is used as the mask; otherwise, grounding is performed.

**Segmentation Head.** For pixel-wise grounding, a training-free method (*i.e.*, text-to-point strategy) is used to unlock LMM's grounding ability. The LMM's textual output is mapped into a point-based prompt, which is processed by a SAM-like segmentation model to generate the final mask.

## 4.2 TEXT-TO-POINT STRATEGY

To leverage textual output to guide the grounding process, we implicitly map the positional terms to the point prompt coordinates. Specifically, with the assess of the logits from the hidden states of LMM, which reflect the probability distribution of every token, we can get the probabilities of positional terms by applying a close-set softmax operation to the corresponding logits. We adopt the term *left* and *right* for horizontal position representation, and *top* and *bottom* for vertical position representation. The softmax operation is illustrated as follows:

$$p_{l_i} = \frac{e^{\chi_{l_i}/\tau}}{\sum_j e^{\chi_{l_j}/\tau}}, \quad l = \{w, h\} \tag{2}$$

$$\{w_i|_{i=0}^1\} = \{left, right\}, \quad \{h_i|_{i=0}^1\} = \{top, bottom\} \tag{3}$$

where $\tau$ denotes the temperature, $\chi$ is the logits. $w$ and $h$ are the collection of horizontal position terms and vertical position terms, respectively. We define the normalized x-coordinate and y-coordinate values of an image from 0 to 1. With the rule that coordinate origin is the top left corner, the terms *left* and *right* can be mapped to value 0 and 1 respectively. Similarly, *top* can be mapped to value 0

and *bottom* can be mapped to value 1. Finally, we calculate the point coordinates using a weighted average and then scale them back to the original size. The process is shown as:

$$x = \sum ip_{w_i} \times W, \quad y = \sum ip_{h_i} \times H \quad (4)$$

where $W$ and $H$ represent the width and height of the original image.

Once we get the point coordinates, we adopt it as prompt for segmentation without more supervision. Unlike the previous works, that generate the prompt explicitly by training the LMM to output the special token, we implicitly convert text into point prompt, keeping the textual output format unaltered. Moreover, compared to other methods (Wu et al., 2024c; Cao et al., 2024) that implicitly generate mask prompt by attention map, which either cause high memory overhead or require an additional image encoder. We bridge the gap of LMM and segmentation model in a simple way and can be adopt widely to any LMM.

### 4.3 Hybrid Dataset Training

To stimulate the model with strong and generalizable quality perception ability, we scale up the training dataset by taking two existing dataset partially with our own dataset. Our training dataset consists of three parts. i) The Q-Instruct(Wu et al., 2024a) dataset, which contains global and local quality question answering, includes the low-level visual perception tasks. ii) The DQ-495K(You et al., 2024b) dataset. A collection of distortion identification tasks and assessment reasoning tasks. It is helpful for enhancing the distortion perception and visual damage description capabilities. iii) Ours dataset, which contains four tasks, inspiring the fine-grained perception ability of the models. We believe that the combination of these datasets contributes to the enhancement of quality-centric understanding abilities and conversational ability, enabling it to unify a wide range of tasks.

**Training Objectives.** The segmentation head is weight-frozen without specific fine-tuning. As for base LMM, we apply the instruction tuning to make the base LMM be quality-aware. Following most supervised fine-tuning works (Yin et al., 2023; Liu et al., 2023b) in LMMs, we apply the next token prediction loss as our training objective, which is a cross-entropy loss for token generation.

## 5 Experiment

### 5.1 Experimental Setup

**Implementation Details.** We use the pretrained Phi-3.5-Vision (Abdin et al., 2024) as our base model. Following previous works(Lai et al., 2024; Rasheed et al., 2024), we adopt SAM as the segmentation head, enabling the grounding ability of the model. In the fine-tuning stage, we apply the parameter-efficient technique LoRA (Hu et al., 2022) for LLM, and apply full-tuning for the visual encoder as well as the visual projector. We adopt the AdamW (Loshchilov & Hutter, 2017) optimizer, setting the initial learning rate of $1 \times 10^{-4}$ with a cosine annealing scheduler. The model is trained for 1 epoch with the batch size of 2.

**Benchmark and Metrics.** We introduce a new benchmark to evaluate four proposed tasks—global description, local description, grounding, and referring (referring$^{short}$, referring$^{long}$)—based on sampled subsets consisting of 629, 460, 586, 780, and 786 instances, respectively. And we divide all the evaluation tasks into three task major categories: text-based answering task, pixel-wise grounding task and visual quality scoring task. For the **text-based answering tasks** (question answering and description), we use Accuracy and GPT Score as evaluation metrics. This benchmark includes both a commonly used question answering dataset (Q-Bench-A1) and our newly constructed benchmark. For the **pixel-wise grounding task**, the ground truth comprises both textual answers and segmentation masks. We adopt the mean Intersection over Union (mIoU) as the evaluation metric, following (Chen et al., 2024a;c). Finally, for the **visual quality scoring task**, we employ the Pearson Linear Correlation Coefficient (PLCC) and the Spearman Rank Correlation Coefficient (SRCC) as evaluation metrics.

### 5.2 Results

**Text-based Answering Tasks: Quantitative Results of Our Benchmark.** As shown in Tab. 1, for description tasks, our model significantly outperforms other models across both global and local descriptions. This highlights our model's superior ability to generate high-quality, detailed descriptions in comparison to existing IQA-specific LMMs. In terms of grounding performance,

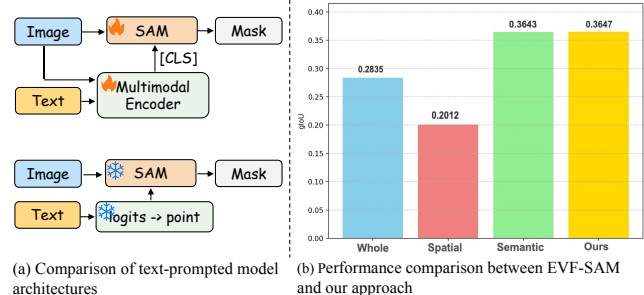

(a) Comparison of text-prompted model architectures

(b) Performance comparison between EVF-SAM and our approach

Figure 5: **Comparison between EVF-SAM and our approach.** (a) Comparison of two text-prompted model architectures. (b) Grounding performance comparison on our dataset.

our model leads the highest score, surpassing all other models in this task. This suggests that our approach effectively integrates the grounding task, providing more accurate region-wise perception ability compared to other methods. In quality referring task, our model performs strongly in both *short* and *long* setting. Although Qwen-2.5-VL performs sub-optimally in grounding, it exhibits poor results in referring, underscoring its weakness in fine-grained region-wise perception. Moreover, we find that the existing IQA-specific LMMs have poor instruction-following ability to the unseen queries, as they have been overfitted to their own dataset. As demonstrated in Fig.4 (a), for local quality description task, all other models produce irrelevant global descriptions that do not align with the query and fail to point out the location of the distortions, yielding for poor fine-grained perception ability. Overall, our model excels in all evaluated tasks, where it achieves superior scores compared to existing models. This indicates the effectiveness of our approach in addressing the challenges of multi-granularity image quality assessment.

**Text-based Answering Tasks: Quantitative Results of Q-Bench.** We evaluate the general low-level perception capability of our model on Q-Bench-A1 in the multiple-choice question answering setting. As shown in Tab. 2, our model outperforms all baselines, indicating comprehensive low-level quality understanding.

Table 1: **Quantitative comparison on our benchmark.** 5 round GPT-4V score is adopted as metric for description and grounding tasks, with a scale of 0-10 and 0-5 respectively. Accuracy is used to evaluate quality referring task.

| Model | Global Des. | Local Des. | Grounding | Ref$^{short}$ | Ref$^{long}$ |
|---|---|---|---|---|---|
| Qwen-2.5-VL(7B) (Bai et al., 2025) | 3.30 | 3.21 | 1.25 | 0.087 | 0.117 |
| Q-Instruct (Wu et al., 2024a) | 3.80 | 3.88 | 0.93 | 0.116 | 0.129 |
| Co-Instruct (Wu et al., 2024b) | 3.50 | 3.53 | 0.52 | 0.121 | 0.119 |
| DepictQA-Wild (You et al., 2024a) | 3.49 | 4.16 | 0.24 | 0.196 | 0.164 |
| Phi-3.5-Vision (Abdin et al., 2024) | 4.03 | 3.01 | 0.68 | 0.100 | 0.101 |
| Ours | **6.17** | **6.55** | **2.29** | **0.321** | **0.290** |

Table 2: **Quantitative comparison on LLVisionQA-dev dataset from Q-Bench-A1.**

| Model | Accuracy |
|---|---|
| random guess | 37.80% |
| LLaVA-v1.5(7B) (Liu et al., 2023a) | 58.66% |
| LLaVA-v1.5(13B) (Liu et al., 2023a) | 62.14% |
| InternLM-XComposer-VL (Zhang et al., 2023a) | 65.35% |
| mPLUG-Owl2 (Ye et al., 2024) | 61.61% |
| Q-Instruct (Wu et al., 2024a) | 67.56% |
| Ours | **70.30%** |

**Pixel-wise Grounding Task: Quantitative Results of Our Benchmark.** We evaluate the pixel-wise quality grounding task using two general LMM-based grounding methods and two quality-aware models: Q-Ground and Q-Instruct. Notably, we adapt Q-Instruct with a text-to-point strategy to activate its grounding capability. Traditional non-LMM segmentation approaches are excluded from our evaluation, as they lack the capacity for the complex and implicit reasoning required by our benchmark. As shown in Tab.3, general LMM-based grounding models exhibit limited performance due to their insufficient quality-aware visual perception under severe distortions and the absence of quality-specific knowledge in the language model. In contrast, our model achieves the best performance, with Q-Instruct ranking second, outperforming both general-purpose and quality-specific grounding methods. Furthermore, the visualization in Fig.4 (b) demonstrates that other models generate incorrect or empty masks, underscoring the superior pixel-level quality grounding capability of our method.

Table 3: **Visual quality grounding evaluation on our benchmark.** The results of Q-Ground are obtained from our own reproduction.

| Dataset | Ours | | | |
|---|---|---|---|---|
| Model | DAO-G | HyD-G | SiD-G | Average |
| LISA(7B) (Lai et al., 2024) | 0.063 | 0.101 | 0.086 | 0.078 |
| Sa2VA(8B) (Yuan et al., 2025) | 0.186 | 0.194 | 0.209 | 0.192 |
| Q-Instruct (Wu et al., 2024a) | 0.243 | 0.315 | 0.229 | 0.264 |
| Q-Ground* (Chen et al., 2024a) | 0.192 | 0.176 | 0.202 | 0.190 |
| Ours | **0.375** | **0.354** | **0.342** | **0.364** |

Table 4: **Ablative study of hybrid datasets training.**

| ID | Ours | Q-Instruct | DQ-495K | Visual Encoder | Global Des. | Local Des. | Grounding | Ref$^{short}$ | Ref$^{long}$ |
|---|---|---|---|---|---|---|---|---|---|
| I | ✓ | | | ✓ | 5.78 | 6.27 | 1.98 | 0.291 | 0.259 |
| II | ✓ | ✓ | | ✓ | 5.95 | 6.31 | 2.00 | 0.301 | 0.250 |
| III | ✓ | ✓ | ✓ | ✓ | 5.95 | 6.36 | 2.07 | 0.316 | 0.282 |
| IV | ✓ | ✓ | ✓ | | 5.64 | 6.27 | 1.94 | 0.275 | 0.234 |
| V | ✓ | ✓ | ✓ | ✓ | **6.17** | **6.55** | **2.29** | **0.321** | **0.290** |

Table 5: **Visual quality grounding evaluation on Q-Ground-Test.**

| Dataset | Q-Ground-Test |
|---|---|
| **Model** | **mIoU** |
| LISA (Lai et al., 2024) | 0.227 |
| PixelLM (Ren et al., 2024) | 0.252 |
| Q-ground (Chen et al., 2024a) | 0.271 |
| Ours | **0.293** |

**Pixel-based Grounding Task: Quantitative Results of Q-Ground-Test.** To evaluate our model on Q-Ground-Test, we use the same prompt format in our training process to prompt the model with ideal output format that contains positional terms for point mapping. We clarify that this does not provide any additional quality-related information. As demonstrated in Tab. 5, while all baseline methods are re-trained or fine-tuned with Q-Ground-100K, our model surpass all of them in a zero-shot manner and without fine-tuning on any grounding dataset. Furthermore, unlike Q-Ground, our model do not involves a quality text reference in the prompt. This further validates the effectiveness of our model.

**Visual Quality Scoring Task.** In addition, we assess the visual scoring ability by testing on two datasets for authentic distortion and synthetic distortion. Results in Tab. 6 show that, despite the absence of task-specific training, the model achieves best performance.

Table 6: **Quantitative comparison over 2 challenging visual scoring datasets.** Except ours and Q-Instruct, all the other models are trained on KonIQ (Hosu et al., 2020) dataset (**SRCC/PLCC**).

| Dataset Type | In-the-wild | Artificial | Average |
|---|---|---|---|
| **Model / Dataset** | **FLIVE** | **KADID-10K** | |
| MUSIQ (Ke et al., 2021) | 0.467/0.565 | 0.556/0.575 | 0.541 |
| CLIP-IQA+ (Wang et al., 2023) | 0.316/0.427 | 0.654/0.653 | 0.513 |
| ManIQA (Yang et al., 2022) | 0.401/0.512 | 0.465/0.499 | 0.469 |
| Q-Instruct (Wu et al., 2024a) | 0.432/0.545 | 0.698/0.676 | 0.588 |
| Ours | **0.439/0.541** | **0.815/0.783** | **0.645** |

## 5.3 ABLATIVE STUDY

**Effectiveness of Hybrid Datasets Training.** We evaluate the effectiveness of hybrid dataset training on our benchmark, as demonstrated in Tab. 4. As for experiment I, we only incorporate our own dataset for training, which results in performance degradation. Experiment II, III and V show that incorporating quality-related tasks can promote performance improvement. Since our dataset mostly consists of synthetically distorted images, the synthetically distorted dataset DQ-495K contributes to more improvement. In experiment IV, we further validate the effectiveness of training the parameters of the visual encoder, since the original visual encoder has limited quality perception capability, keeping it frozen results in poor performance.

**Effectiveness of Text-to-point Strategy.** As for text-prompted SAM architectures, we compare our text-to-point strategy with EVF-SAM (Zhang et al., 2024), which links the text and the segmentation model(*i.e.*, SAM) by adopting a trainable multimodal encoder to generate prompt. The comparison of the model architectures is illustrated in Fig. 5 (a). Our model's textual output is structured in two parts: a spatial answer first and a semantic answer second. Based on this structure, we evaluate EVF-SAM using three types of textual inputs: the *whole* answer, the *spatial* component only, and the *semantic* component only. As shown in Fig. 5 (b), EVF-SAM achieves its best performance when prompted with the semantic text alone, and its performance degrades when using the whole answer. Despite EVF-SAM being fine-tuned jointly with the multimodal encoder and SAM, our zero-shot text-to-point strategy still achieves slightly better results. These results validate both the effectiveness and efficiency of our proposed grounding strategy.

## 6 CONCLUSION

In this study, we introduce a novel four-task paradigm emphasizing region-wise quality understanding. To support it, we construct a unified multi-task dataset, IQA-Octopus-33K, encompassing global and local description, pixel-wise grounding and region-wise referring, enabling multi-granularity perception. Furthermore, we present the IQA-Octopus framework, which adopts a conflict-free two-stage optimization strategy: it first optimizes text-based perception tasks, then transitions to pixel-level perception through the text-to-point grounding strategy. Our work establishes a new benchmark for fine-grained quality assessment tasks and holds promise for broader applications in quality-related domains.

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

# A APPENDIX

## A.1 LLM USEAGE

Large Language Models (LLMs) were used solely to improve the manuscript's language, readability, and clarity, including sentence rephrasing and grammar checking. The LLM did not contribute to the research design, methodology, or scientific content. All ideas, analyses, and results were developed by the authors, who take full responsibility for the manuscript.

## A.2 DATASET DETAILS

**Multi-distortion accumulation setting.** In scenarios involving multiple distortions, the accumulation order becomes unidentifiable when different orders produce identical visual outcomes. To make up for this, we manually define the distinguishable distortion accumulation orders. Following DQ-495K (You et al., 2024b), we first determine the distortion combinations by removing those exhibiting similar or contradictory effects, ensuring the distortion categories clear and distinguishable. Secondly, we enumerate all possible orders and filter the recognizable ones, which is shown in Tab. 8.

Table 7: **Statistics of four tasks in our dataset.**

| Tasks | Global Des. | Local Des. | Grounding | Ref$^{short}$ | Ref$^{long}$ |
|---|---|---|---|---|---|
| Train | 3251 | 9742 | 11669 | 4506 | 4222 |
| Test | 629 | 460 | 586 | 780 | 786 |

**Details of Dataset Construction.** We construct synthetically distorted images by using the pristine images from Kadis-700K (Lin et al., 2019), and we leverage human to annotate the authentically distorted images from KonIQ (Hosu et al., 2020). The image source statistics are illustrated in Fig. 6. For synthetic distortions, we use the distortions categories from DQ-495K (You et al., 2024b). For human annotations, the visualized wordcloud of distortion categories is shown in Fig. 7. Generally, the statistics of four tasks in our dataset in demonstrated in Tab. 7. All the question templates are shown in Tab. 9, Tab. 10, Tab. 11, Tab. 12, Tab. 13, Tab. 14, Tab. 15, Tab. 16 and Tab. 17.

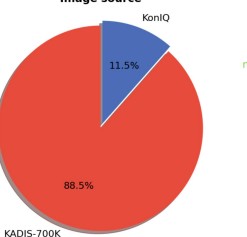

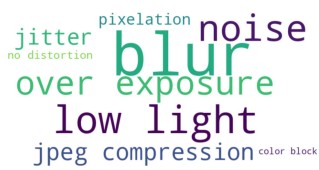

Figure 7: **Wordcloud map of human annotated distortion categories.**

Figure 6: **Image source statistics.**

Table 8: **Distinguishable distortion accumulation orders.**

| First Distortion | All Possible Second Distortions |
|---|---|
| Blur | Compression, Noise |
| Compression | Blur, Noise |
| Contrast Weaken | Noise, Blur, Compression, Contrast Weaken, Pixelate, Saturate Weaken |
| Pixelate | Noise |
| Saturate Weaken | Noise |

## A.3 MORE RESULTS

As the visualized results of local quality description and visual quality grounding are illustrated in the main manuscript, we demonstrated more qualitative results of global quality description and visual

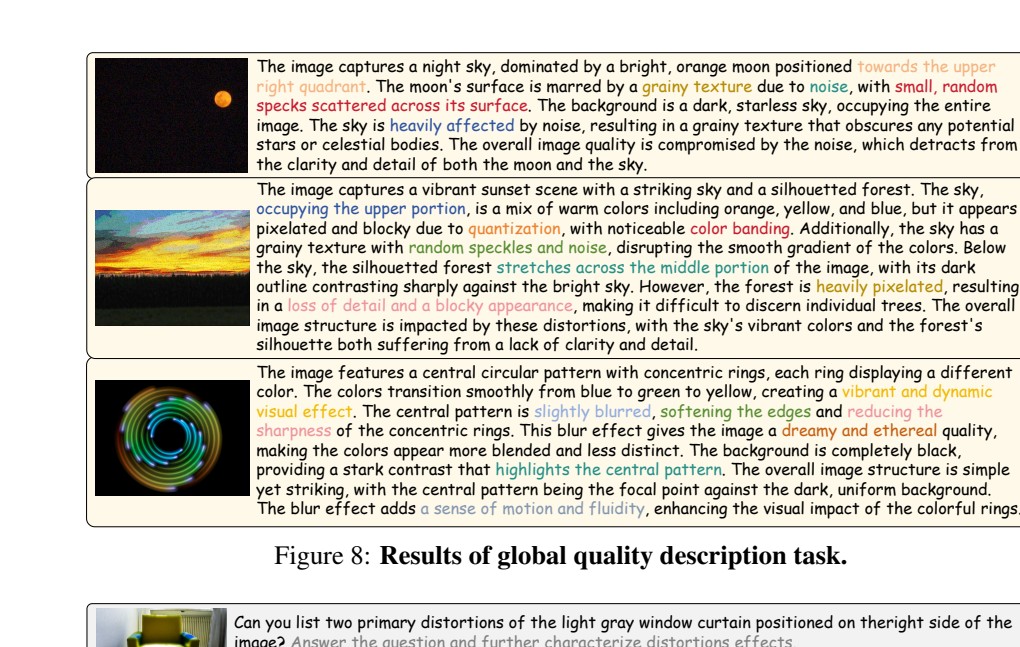

Figure 8: **Results of global quality description task.**

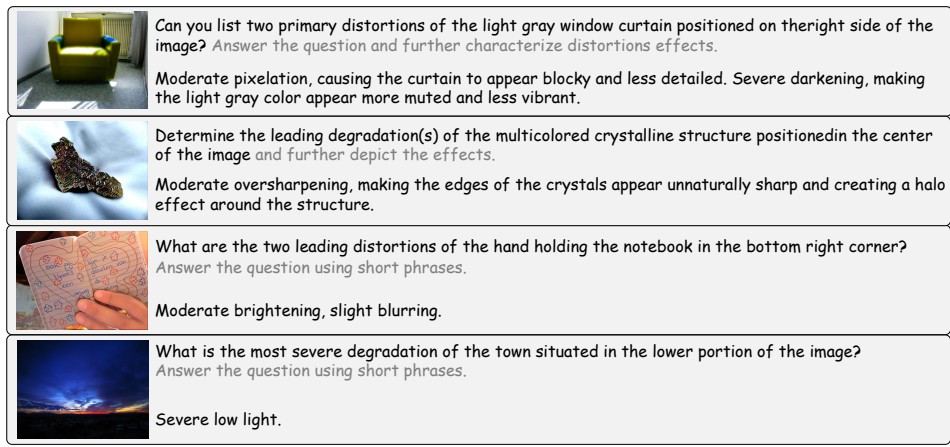

Figure 9: **Results of Visual Quality Referring task.**

quality referring in Fig. 8 and Fig. 9. In global quality description task, our model output is more detailed, which can describe the distortion type and visual effect with fine-grained spatial location. For visual quality referring task, our model not only identifies distortions at the region level but also explains their visual impact, showing its strong region-level quality understanding capability. These results further validate the comprehensive multi-granularity perception capability of our model.

Table 9: **Question pool for local quality description task.**

| # | Question |
|---|---|
| 1 | Conduct a detailed description about {} with particular attention to quality evaluation. |
| 2 | Deliver a thorough description of {} with an emphasis on quality assessment. |
| 3 | Offer a detailed analysis of {} with a focus on evaluating its quality. |
| 4 | Supply a thorough analysis of {} emphasizing quality assessment. |
| 5 | Conduct a detailed analysis of {} that prioritizes quality evaluation. |
| 6 | Create a complete evaluation of {} with particular attention to quality. |
| 7 | Conduct a thorough evaluation of {} that prioritizes quality insights. |
| 8 | Provide a careful evaluation of {} with attention to quality assessment. |
| 9 | Create a meticulous review of {} with a focus on its quality aspects. |
| 10 | Write a thorough assessment of {} that underscores quality evaluation. |
| 11 | Conduct a thorough analysis of {} that highlights quality assessment. |
| 12 | Present an extensive evaluation of {} with a focus on the aspects of quality. |
| 13 | Conduct an in-depth appraisal of {} that considers quality-related factors. |
| 14 | Furnish a detailed evaluation of {} with insights on quality-related factors. |
| 15 | Can you offer an in-depth description of {} that highlights quality evaluation? |
| 16 | How would you characterize {} while focusing on quality evaluation? |
| 18 | Could you provide a description of {} that highlights its quality aspects? |

Table 10: **Question pool for global quality description task.**

| # | Question |
|---|----------|
| 1 | Assess the quality of the image with a detailed explanation. |
| 2 | Analyze the image quality and provide a thorough explanation. |
| 3 | Examine the quality of the image with an in-depth discussion. |
| 4 | Evaluate the quality of the image with a comprehensive analysis. |
| 5 | Provide an evaluation of the image quality with a complete explanation. |
| 6 | Appraise the quality of the image with a comprehensive overview. |
| 7 | Deliver a thorough evaluation of the quality of the image, highlighting both strengths and weaknesses. |
| 8 | Present an in-depth analysis of the quality of the image, addressing its advantages and areas needing improvement. |
| 9 | Offer a detailed assessment of the image quality, encompassing both positive aspects and opportunities for enhancement. |
| 10 | Conduct a comprehensive review of the quality of the image, noting strengths as well as potential improvements. |
| 11 | Present a detailed analysis of the image quality, focusing on its strong points and areas that could be enhanced. |
| 12 | Present a complete assessment of the image quality, addressing its advantages and identifying areas for improvement. |
| 13 | Offer a thorough analysis of the image quality, mentioning both favorable aspects and areas to be enhanced. |
| 14 | Present a holistic assessment of the quality of the image, detailing its strengths and areas that require enhancement. |
| 15 | Conduct a detailed review of the quality of the image, focusing on both its strong features and areas that could benefit from refinement |
| 16 | Deliver a comprehensive analysis of the quality of the image, concentrating on its merits and areas needing improvement |
| 17 | Investigate the quality of the image while considering aspects that contribute to its degradation. |
| 18 | Consider the factors affecting clarity as you assess the quality of the image. |
| 19 | Analyze the quality of the image while examining factors that lead to its degradation. |
| 20 | Investigate the image quality while evaluating the factors that result in its degradation. |
| 21 | How do you assess the quality of the image, and what aspects contribute to your opinion? |
| 22 | What are your thoughts on the quality of the image? Please elaborate on your perspective. |
| 23 | How would you evaluate the quality of the image? Share a detailed explanation of your opinion. |
| 24 | What is your perspective on the quality of the image? Expand on your evaluation. |
| 25 | Can you deliver an in-depth evaluation of the quality of the image? |
| 26 | Could you conduct a complete evaluation of the quality of the image? |

Table 11: **Question pool for hybrid distortion intensity grounding task.**

| # | Question |
|---|---|
| 1 | which is the most degraded region in the evaluated image? |
| 2 | Which region exhibits the highest impact from distortions among all the regions in the evaluated image? |
| 3 | Which region shows the most severe degradation across all the regions in the evaluated image? |
| 4 | Which region suffers the most degradation compared to the other regions in the evaluated image? |
| 5 | Among all the regions in the evaluated image, which is the most significantly degraded one? |
| 6 | Which region has the lowest quality among all the regions due to distortions in the evaluated image? |
| 7 | Which region exhibits the lowest quality as a result of distortions in the evaluated image? |
| 8 | What region shows the greatest decline in quality because of distortions in the evaluated image? |
| 9 | Which region is marked by the poorest quality due to distortions in the examined image? |
| 10 | Pinpoint the region with the worst distortion in the evaluated image. |
| 11 | Determine the region that is most degraded in the evaluated image. |
| 12 | Identify the region with the highest level of distortion in the evaluated image. |
| 13 | which is the least degraded region in the evaluated image? |
| 14 | Which region is least affected by distortions compared to others regions in the evaluated image? |
| 15 | Which region has experienced the minimal level of degradation in the evaluated image? |
| 16 | Which region exhibits the lowest degree of degradation compared to all the other regions in the evaluated image? |
| 17 | Among all regions in the evaluated image, which one has the best quality with minimal distortion influence? |
| 18 | What is the region with the highest quality and minimal distortion effects in the evaluated image? |
| 19 | Which region has the highest quality with minimal impact from distortion in the evaluated image? |
| 20 | In terms of quality, which region is the least affected by distortion in the evaluated image? |
| 21 | Which region demonstrates the best quality in the evaluated image? |
| 22 | Find the region that shows the least amount of degradation in the evaluated image. |
| 23 | Determine the region that is least degraded in the evaluated image. |
| 24 | Identify the region with the lowest level of distortion in the evaluated image. |

Table 12: **Question pool for single distortion intensity grounding task.**

| # | Question |
|---|---|
| 1 | Which region shows the highest level of {} in the evaluated image? |
| 2 | Which region shows the most severe {} in the evaluated image? |
| 3 | What region has the highest amount of {} in the evaluated image? |
| 4 | Which region experiences the highest degree of {} in the evaluated image? |
| 5 | Which region has the greatest level of {} in the evaluated image? |
| 6 | Pinpoint the region characterized by the highest level of {} in the evaluated image. |
| 7 | Determine the region with the highest intensity of {} in the evaluated image. |
| 8 | Which region has the most substantial {} effect in the evaluated image. |
| 9 | What region exhibits the least amount of {} in the evaluated image? |
| 10 | What is the region with the minimal level of {} in the evaluated image? |
| 11 | Which region exhibits the lowest degree of {} in the evaluated image? |
| 12 | Which region ranks the lowest in terms of {} in the evaluated image? |
| 13 | Which region demonstrates the least extent of {} in the evaluated image? |
| 14 | Identify the region that has the minimal {} in the evaluated image. |
| 15 | Which region has the most negligible {} effect? |
| 16 | Determine the region with the lowest intensity of {} in the evaluated image. |

Table 13: **Question pool for distortion accumulation order grounding task.**

| # | Question |
|---|----------|
| 1 | Which region follows the distortion addition sequence of {} and {} in the evaluated image? |
| 2 | Which region add {} first, followed by {} in distortion addition process in the evaluated image? |
| 3 | Identify the region that follows the {}-first, {}-second distortion addition pattern in the evaluated image. |
| 4 | What region follows the pattern of adding {} before {} in the evaluated image? |
| 5 | Determine the region that corresponds to the distortion addition order of {}, then {} in the evaluated image. |
| 6 | Which region matches the pattern of distortion addition that begins with {} and ends with {} in the evaluated image? |
| 7 | Which region begins the distortion addition process with {} in the evaluated image? |
| 8 | Which region adds {} at the beginning of the distortion sequence in the evaluated image? |
| 9 | Which region integrates {} first during the distortion addition process in the evaluated image? |
| 10 | Identify the region that brings in {} first in the distortion addition process in the evaluated image. |
| 11 | What region initiates the distortion sequence with the addition of {} in the evaluated image? |
| 12 | Determine the region that includes {} first in the distortion addition process within the evaluated image. |
| 13 | Which region integrates {} last during the distortion addition process in the evaluated image? |
| 14 | Which region incorporates {} as the last element in the distortion addition process in the evaluated image? |
| 15 | What region of the evaluated image adds {} as the final step in the distortion sequence? |
| 16 | Which region adds {} at the end of the distortion sequence in the evaluated image? |
| 17 | What region finishes the distortion sequence by adding {} in the evaluated image? |
| 18 | Determine the region that includes {} last in the distortion addition process within the evaluated image. |

Table 14: **Question pool for visual quality referring task** about single distortion in the short answer setting.

| # | Question |
|---|----------|
| 1 | Identify the most critical one distortion of {}. Answer the question using short phrases. |
| 2 | Pinpoint the foremost image quality issue in the evaluated image. Answer the question using short phrases. |
| 3 | List the most significant distortion related to {}. Answer the question using short phrases. |
| 4 | Can you list one primary distortion of {}? Answer the question using short phrases. |
| 5 | What is the leading distortion of {}? Answer the question using short phrases. |
| 6 | In terms of image quality, what is the most glaring issue of {}? Answer the question using short phrases. |
| 7 | What is the most severe degradation of {}? Answer the question using short phrases. |
| 8 | Pinpoint the foremost image quality issue(s) of {}. Answer the question using short phrases. |
| 9 | What distortion(s) most detrimentally affect the overall quality of {}? Answer the question using short phrases. |
| 10 | What distortion(s) are most prominent when examining {}? Answer the question using short phrases. |
| 11 | What distortion(s) are most apparent of {}? Answer the question using short phrases. |
| 12 | What distortion(s) stand out of {}? Answer the question using short phrases. |
| 13 | Identify the most critical distortion(s) of {}. Answer the question using short phrases. |
| 14 | Determine the leading degradation(s) of {}. Answer the question using short phrases. |

Table 15: **Question pool for visual quality referring task** about single distortion in the long answer setting.

| # | Question |
|---|----------|
| 1 | Identify the most critical distortion of {} and depict its effects. |
| 2 | Pinpoint the foremost image quality issue in the evaluated image and elaborate on its effects. |
| 3 | List the most significant distortion related to {} and describe its effects. |
| 4 | Can you list one primary distortion of {} and detail its effects? |
| 5 | What is the leading distortion of {}? Answer the question and describe its effects. |
| 6 | In terms of image quality, what is the most glaring issue of {}? Answer the question and elaborate on its effects. |
| 7 | What is the most severe degradation of {}? Answer the question and characterize its effects. |
| 8 | Pinpoint the foremost image quality issue(s) of {} and describe its effects. |
| 9 | What distortion(s) most detrimentally affect the overall quality of {}? Answer the question and elaborate on its effects. |
| 10 | What distortion(s) are most prominent when examining {}? Answer the question and depict its effects. |
| 11 | What distortion(s) are most apparent of {}? Answer the question and detail its effects. |
| 12 | What distortion(s) stand out of {}? Answer the question and describe its effects. |
| 13 | Identify the most critical distortion(s) of {} and elaborate on its effects.Identify the most critical distortion(s) of {} and elaborate on its effects. |
| 14 | Determine the leading degradation(s) of {} and depict its effects. |

Table 16: **Question pool for visual quality referring task** about multi-distortions in the short answer setting.

| # | Question |
|---|----------|
| 1 | Identify two most critical distortions of {}. Answer the question using short phrases. |
| 2 | Pinpoint two foremost image quality issues in the evaluated image. Answer the question using short phrases. |
| 3 | List two most significant distortions related to {}. Answer the question using short phrases. |
| 4 | Can you list two primary distortions of {}? Answer the question using short phrases. |
| 5 | What are the two leading distortions of {}? Answer the question using short phrases. |
| 6 | In terms of image quality, what are the two most glaring issues of {}? Answer the question using short phrases. |
| 7 | What are the two most severe degradations of {}? Answer the question using short phrases. |
| 8 | Pinpoint the foremost image quality issue(s) of {}. Answer the question using short phrases. |
| 9 | What distortion(s) most detrimentally affect the overall quality of {}? Answer the question using short phrases. |
| 10 | What distortion(s) are most prominent when examining {}? Answer the question using short phrases. |
| 11 | What distortion(s) are most apparent of {}? Answer the question using short phrases. |
| 12 | What distortion(s) stand out of {}? Answer the question using short phrases. |
| 13 | Identify the most critical distortion(s) of {}. Answer the question using short phrases. |
| 14 | Determine the leading degradation(s) of {}. Answer the question using short phrases. |

Table 17: **Question pool for visual quality referring task** about multi-distortions in the long answer setting.

| # | Question |
|---|----------|
| 1 | Identify two most critical distortions of {} and describe their effects. |
| 2 | Pinpoint two foremost image quality issues in the evaluated image and describe their effects. |
| 3 | List two most significant distortions related to {} and describe their effects. |
| 4 | Can you list two primary distortions of {}? Answer the question and characterize their effects. |
| 5 | What are the two leading distortions of {}? Answer the question and explain their effects. |
| 6 | In terms of image quality, what are the two most glaring issues of {}? Answer the question and elaborate on their effects. |
| 7 | What are the two most severe degradations of {}? Answer the question and elaborate on their effects. |
| 8 | Pinpoint the foremost image quality issue(s) of {} and depict their effects. |
| 9 | What distortion(s) most detrimentally affect the overall quality of {}? Answer the question and detail their effects. |
| 10 | What distortion(s) are most prominent when examining {}? Answer the question and describe their effects. |
| 11 | What distortion(s) are most apparent of {}? Answer the question and depict their effects. |
| 12 | What distortion(s) stand out of {}? Answer the question and describe their effects. |
| 13 | Identify the most critical distortion(s) of {} and detail their effects. |
| 14 | Determine the leading degradation(s) of {} and describe their effects. |

