# OpenReview forum: "IQA-Octopus: Unified Multi-Granularity Image Quality Assessment with Reasoning, Grounding and Referring"
_ICLR.cc/2026/Conference — Submitted to ICLR 2026_

### Official Review · Reviewer_pwEe · 2025-10-21

**Soundness:** 2
**Presentation:** 3
**Contribution:** 2
**Rating:** 4
**Confidence:** 5

**Summary:**

This paper introduces IQA-Octopus, a unified image quality assessment (IQA) framework that integrates reasoning, grounding, and referring within a single large multimodal model (LMM) architecture. Unlike prior works that focus on isolated dimensions of visual quality (e.g., global description, pixel-wise grounding), IQA-Octopus emphasizes multi-granularity perception by jointly modeling global, local, and pixel-level understanding.

**Strengths:**

1. First LMM-based IQA model combining reasoning, grounding, and referring, addressing multi-granularity perception in a coherent architecture.

2. The text-to-point method avoids retraining segmentation heads while maintaining reasoning ability.

3. High-quality dataset: IQA-Octopus-33K integrates both synthetic and real distortions, enabling comprehensive evaluation.

**Weaknesses:**

1. The proposed IQA-Octopus-33K dataset, while diverse, contains only 33K samples. This scale may be insufficient for robust instruction-tuning at the LMM level, and expanding the dataset could further strengthen the empirical claims.

2. The evaluation is restricted to images. Assessing the model’s generalization to video or 3D quality tasks would better demonstrate the versatility of the proposed multi-granularity reasoning framework.

3. Although the hybrid dataset ablation is insightful, the contribution of each sub-task (reasoning, grounding, referring) is not explicitly disentangled, leaving uncertainty about their respective roles in overall performance gains.

4. The combination of SAM-based grounding and LoRA tuning increases architectural complexity, yet runtime, computational cost, and efficiency trade-offs are not discussed.

5. The dataset generation partially relies on InternVL outputs, which may introduce variability. Full release of generation templates and code would be necessary to ensure reproducibility and transparency.

**Questions:**

1. Do the authors plan to expand IQA-Octopus-33K or incorporate additional data sources to improve instruction-tuning robustness and cross-domain generalization?

2. Could the proposed framework be extended or tested on video or 3D quality assessment tasks to validate its applicability beyond static images?

3.  Can the authors provide a finer-grained ablation that isolates the contributions of reasoning, grounding, and referring sub-tasks to better understand their individual impacts?

4. What is the inference-time overhead of integrating SAM and LoRA compared to a standard LMM baseline, and how does this affect scalability in real-world scenarios?

5. How do the authors ensure consistent data generation from InternVL, and will they release the corresponding scripts and templates to enable full reproducibility?

---

### Official Review · Reviewer_suQi · 2025-10-26

**Soundness:** 2
**Presentation:** 3
**Contribution:** 3
**Rating:** 4
**Confidence:** 4

**Summary:**

This paper proposes the IQA-Octopus framework to unify multi-granularity image quality assessment tasks, including reasoning, localization (grounding), and referring. The authors construct the IQA-Octopus-33K dataset, which covers four task paradigms: global/local quality description, pixel-level quality grounding, and region-level quality referring (Section 3, Fig. 1). The core methodological contributions are: (1) an LMM based on Phi-3.5-Vision is instruction-tuned on a mixed dataset (the proposed 33K data, Q-Instruct, and DQ-495K) to learn multi-granularity reasoning and referring at the text level (Section 4.3); and (2) a “text-to-point” strategy that, at inference time, extracts the probability distribution of location words (top/bottom/left/right) from the LMM’s logits, then produces a single coordinate via weighted averaging (Eqs. 2-4), which serves as a zero-shot prompt to drive a frozen SAM for pixel-level segmentation (Section 4.2). Experimental results show competitive performance on both the authors’ in-house benchmark (Table 1) and external benchmarks (Tables 2 and 5). Notably, on Q-Ground-Test (Table 5), the model achieves 0.293 mIoU in a zero-shot setting, surpassing Q-Ground (0.271), which requires fine-tuning.

**Strengths:**

The paper’s main strengths lie in its clear motivation and the novelty of the “text-to-point” strategy. The authors explicitly argue that existing methods (e.g., Q-Ground, LISA), which introduce special tokens (such as `\<seg\>`) for explicit localization, “damage instruction-following behavior and reasoning processes” (Section 1, lines 107–109), and they propose to avoid this by implicitly mapping text logits to coordinate points (Section 4.2). This design is technically elegant in that it preserves the integrity of the LMM’s text output (“keeping the text output stainless,” Section 4.2, lines 325–327), avoids adding special tokens to the vocabulary, and does not require an additional segmentation head to be fine-tuned. The ablation study (Fig. 5) strongly supports the effectiveness of this strategy: its zero-shot performance (0.364 mIoU) slightly exceeds EVF-SAM (≈0.35 mIoU), which requires joint training of the multimodal encoder and SAM. In addition, the IQA-Octopus-33K dataset (Section 3) unifies four task paradigms and combines synthetic and real distortions (Fig. 2), providing a more comprehensive benchmark for multi-granularity IQA; its automated annotation pipeline (Section 3.2) also enhances the scalability of data construction.

**Weaknesses:**

The core weakness is the lack of direct empirical validation for key assumptions. First, the paper repeatedly asserts that avoiding special tokens prevents “damaging instruction-following and reasoning ability” (Section 1, lines 107-109; Section 2, lines 123-125), yet provides no ablation study directly substantiating this claim, as the authors do not compare their “text-to-point” method against a baseline trained with explicit localization using special tokens (on the same data and backbone) on pure text reasoning tasks (e.g., Q-Bench-A1 in Table 2). As a result, the central “conflict-free” claim lacks empirical support. Second, a fundamental assumption is that a single point derived via weighted averaging of logits over four location words (Eq. 4) suffices to prompt SAM to segment arbitrary distortion regions (Section 4.2). However, the paper does not discuss or validate this assumption in more complex scenarios. For instance, when distortion comprises multiple disjoint regions (e.g., noise at all four corners) or complex shapes (e.g., annular distortions), the DAO-G task’s mIoU of only 0.375 in Table 3 may hint at this limitation, but no analysis is provided. Finally, in data construction (Section 3.2, Eq. 1), the authors use bounding-box centers to derive location-word labels, effectively establishing a strong association between center points and location words at the training-data level. This introduces conceptual ambiguity with the “zero-shot” grounding claimed in Section 4.2. Is the model merely reproducing this training-time mapping, rather than exhibiting genuine zero-shot generalization?

**Questions:**

On validating the core assumption: A central motivation is the claim that using special tokens “damages instruction-following behavior” (Section 1, lines 107-109). Could the authors provide a direct ablation study comparing IQA-Octopus with a variant trained for explicit localization using special tokens (e.g., the `\<seg\>` mechanism in Q-Ground or LISA), and then evaluate the two models on pure text reasoning benchmarks (e.g., Q-Bench-A1 in Table 2) and dialog-based quality assessment? This is critical to substantiate the “conflict-free” core claim.

On the applicability limits of the single-point prompting strategy: The method in Section 4.2 generates a coordinate via weighted averaging (Eq. 4) as a prompt for SAM. How does this single-point mechanism handle more complex localization scenarios? For example, when distortion consists of multiple non-adjacent regions (e.g., noise in all four corners), or when distortion has complex shapes (e.g., hollow ring-shaped distortions), does a single-point prompt remain effective? Do the mIoU differences across subtasks in Table 3 (DAO-G: 0.375 vs. HyD-G: 0.354) reflect this limitation?

On the nature of “zero-shot” grounding: During data construction (Section 3.2, Eq. 1), bounding-box centers are used to generate location-word (top/bottom/left/right) labels, implying that the training data already encodes an explicit mapping between center coordinates and location words. Does the “zero-shot” grounding in Section 4.2 merely reproduce this training-time mapping rather than demonstrate true zero-shot generalization? How do the authors distinguish between these two cases?

On generalization to real-world distortions: Table 6 shows SOTA performance on the synthetic distortion dataset KADID-10K (0.815/0.783 SRCC/PLCC), but performance on the real-world dataset FLIVE (0.439/0.541) is nearly on par with Q-Instruct (0.432/0.545). Given that Fig. 6 indicates the training data are primarily sourced from the synthetic-distortion KADIS-700K, does this suggest that the model’s fine-grained perception abilities (e.g., grounding and referring as in Tables 1 and 3) rely heavily on patterns specific to synthetic data, thereby struggling to generalize to diverse, atypical real-world distortions? How do the authors explain this performance gap?

---

### Official Review · Reviewer_9AQu · 2025-10-28

**Soundness:** 3
**Presentation:** 2
**Contribution:** 2
**Rating:** 4
**Confidence:** 5

**Summary:**

This paper presents a four-task united IQA framework that can handle multiple tasks, including reasoning, grounding, and referring capabilities. The authors construct the dataset for multi-granularity perception and design a conflict-free two-stage optimization strategy. Additionally, extensive experiments on self-built and public benchmarks demonstrate the framework's effectiveness in text-based answering, pixel-wise grounding, and visual quality scoring tasks.

**Strengths:**

1. The authors propose a unified IQA framework that integrates reasoning, grounding, and referring. The conflict-free two-stage optimization and text-to-point strategy effectively balance multi-granularity perception without compromising reasoning ability.
2. The IQA-Octopus-33K dataset is comprehensive and scalable. The dataset covers four task paradigms, and its construction pipeline introduces SAM tools and InternVL-2.5 to ensure scalability and transparency.
3. The experiments apply various benchmarks covering text-based answering, pixel-wise grounding, and visual quality scoring. Ablation studies on hybrid dataset training and text-to-point strategy also validate key design effectiveness.

**Weaknesses:**

1. Figure 3 with the model framework does not detail how the LMM backbone interacts with the SAM segmentation head, especially in the SAM part. Key components like the "text-to-point conversion module" are missing.
2. In the Image Collection part of Section 3.2, the author mentioned that they employed human annotators to provide reliable ground-truth annotations. However,
3. A minor suggestion is to show the usage of color in figure examples of exhibiting text outputs for different methods(Fig. 1,3,4), and distinguish these methods more clearly(Fig. 5).

**Questions:**

1. Please clarify the dataset construction details. For authentic distortion annotation in KonIQ, the relationship between human annotators and reliable image quality, as well as the inter-annotator agreement for distortion type/region labeling, should be proven.
2. Failure cases are needed for deeper analysis.

---

### Official Review · Reviewer_ktKc · 2025-10-28

**Soundness:** 3
**Presentation:** 4
**Contribution:** 3
**Rating:** 4
**Confidence:** 5

**Summary:**

This paper introduces IQA-Octopus, a novel framework for IQA built upon LMMs. The primary contribution is unifying multi-granularity perception tasks: reasoning, grounding, and referring. To address the lack of suitable training data, the authors create and introduce a new dataset, IQA-Octopus-33K. The proposed method utilizes a two-stage optimization strategy. The first stage trains the model for text-based reasoning and referring tasks. The second stage enables pixel-wise perception using a novel, training-free "text-to-point" strategy. This strategy implicitly converts text logits into point coordinates for a segmentation model (i.e., SAM) , avoiding special tokens that can degrade the LMM's reasoning abilities. The authors demonstrate that their model achieves comparable or state-of-the-art performance across multiple IQA benchmarks

**Strengths:**

- The paper is clearly structured, well written and presented in a clear manner.
- The paper proposes a unified framework that combines multi-granularity reasoning, grounding, and referring for IQA. This is a promising direction, as it aligns with the growing demand for more explainable and comprehensive IQA methods.
- The model demonstrates strong, comparable performance across different tasks.
- The architecture is straightforward, consisting of a standard LMM backbone and a frozen segmentation head. Such simplicity is often beneficial for both interpretability and ease of implementation.

**Weaknesses:**

- Limitation of dataset.
    - Scale: The newly proposed IQA-Octopus-33K dataset  seems limited in scale. With only ~33K total samples (and fewer per specific task, as shown in Table 7 ), it may be insufficient for robustly training an LMM to handle four distinct and complex task paradigms. This concern is amplified by the fact that a large portion of the data is generated automatically (synthetic data pipeline ) or semi-automatically (using InternVL-2.5 for Q&A generation ), which can lead to a lack of diversity and potential propagation of errors from the generator model.
    - Annotation Quality: The reliance on a single open-source model (InternVL-2.5)  for generating all Q&A pairs is a significant concern. The paper's justification for avoiding closed-source models like GPT-4V on the grounds of cost and transparency is weak; generating 33K annotations is not prohibitively expensive, and using an open-source model does not inherently make the data generation process more transparent or guarantee higher quality. Relying on a single model risks baking its specific biases and stylistic into the dataset, potentially limiting the trained model's generalizability.
    - Sufficiency: The paper does not adequately demonstrate that the 33K dataset is sufficient for achieving the reported performance. The ablation study in Table 4 actually suggests the contrary: the model trained only on the IQA-Octopus-33K dataset performs worse than the model trained with additional datasets. A more convincing study is needed to justify that this dataset size is sufficient.

- Method design.
    - Naive Segmentation Prompting: The "text-to-point" strategy, while simple, appears overly naive. It generates only a single point coordinate based on a weighted average of positional term logits. It is known that SAM's performance with a single point prompt can be inaccurate, especially for non-salient objects (or in this case, distortion regions). The paper provides insufficient evidence that this single-point-prompting is robust for accurately segmenting diverse and complex quality degradations.
     - *[Suggestion] A better practice might be splitting the image into smaller grids and predict multiple points to prompt SAM.*

- Results comparison.
    - Unclear Baseline Training (Table 1): The experimental setup for Table 1 is unclear and potentially unfair. The paper does not state whether the baseline models were fine-tuned on the new IQA-Octopus-33K training set or evaluated in a zero-shot manner. Given the specialized task formats, a zero-shot evaluation would likely fail, making the comparison weak. If they were retrained, the drastically lower performance of other methods especially on global desc is surprising and requires explanation.
    - Lack of Backbone Ablation (Table 2): The comparison in Table 2 lacks a critical baseline for fair assessment. As shown in Q-Instruct, the choice of LMM backbone can significantly impact performance. The paper should include results using a similar LMM backbone (e.g., InternVL-2.5) without the proposed method to isolate the contribution of the architecture itself.

In conclusion, this paper is well written and proposes a novel unified framework for multi-granularity image quality assessment, which are potentially useful for several applications. However, there are still several concerns about the dataset, the method design and results comparison.

Therefore, I would currently recommend marginal below acceptance, and would like to see the authors address the above concerns in the rebuttal for the final decision.

**Questions:**

Please see weakness points above.

---

### Meta-Review · Area_Chair_EhZA · 2026-01-06

**Summary:**

The submission proposes IQA-Octopus, aiming to unify reasoning, grounding, and referring for image quality assessment, with a training-free text-to-point mechanism to connect an LMM to SAM. However, reviewers consistently raised concerns about dataset sufficiency/quality, method clarity, and evaluation validity/fairness, leading to uniformly marginal-below-threshold recommendations. Therefore, I decide to reject this paper.

**Reviewer Concerns:**

1) Dataset Scale: The proposed IQA-Octopus-33K is considered too small for robust instruction tuning across multiple task paradigms

2) Method Description and Technical Clarity: The framework figure/description does not clearly explain how the LMM interacts with the SAM head, particularly the SAM part.

**Reviewer Scores:**

Had the reviewer been able to fully participate in the discussion, I believe their score would likely have remained largely unchanged. I appreciate the feedback provided and will carefully address these points in a revised version of the manuscript.

---

### Decision · Program_Chairs · 2026-01-26

Reject